# DIFFSTROKE: HIGH-QUALITY MASK-FREE IMAGE MANIPULATION WITH PARTIAL SKETCHES

## ABSTRACT

Sketches offer a simple yet powerful way to represent object configurations, making them ideal for local image structure manipulation. Traditional methods often treat sketch-based editing as an image inpainting task, requiring both user-provided strokes and masks, which hinders the user experience. Although recent mask-free stroke-based editing methods are more convenient, they often produce significant artifacts or unintentionally modify irrelevant regions. To overcome these challenges, we propose DiffStroke, a mask-free method for high-quality image editing using only partial sketches. Trainable plug-and-play Image-Stroke Fusion (ISF) modules and an effective mask estimator are developed to address the limitations of previous conditional control diffusion models in preserving style consistency and protecting irrelevant areas. The ISF modules fuse stroke encodings with source image features as input conditions, enabling DiffStroke to control local shapes while preserving overall style consistency. The mask estimator automatically predicts masks to preserve irrelevant regions without the need for manual input. Specifically, DiffStroke blends the estimated clean latent image with the encoded source image using the predicted mask, with the mask estimator trained to minimize the error between the blended result and the latent target image. Experimental results on natural and facial images demonstrate that DiffStroke outperforms previous methods in both simple and complex stroke-based image editing tasks.

## 1 INTRODUCTION

Sketching is a widely used, convenient method to convey messages. In particular, it has the advantage of conveying abstract geometric concepts. For example, it is challenging to accurately convey the contours of an item by words, but a sketch can effectively represent contours according to the shape of an object with a minimal number of strokes. Consequently, it is frequently employed as a control condition to direct image generation (Isola et al., 2017; Koley et al., 2023). Thanks to the powerful generative capabilities of the advanced modeling paradigm (Goodfellow et al., 2014; Sohl-Dickstein et al., 2015; Ho et al., 2020), recent work has succeeded in synthesizing realistic images while maintaining the corresponding reference structures (Chen & Hays, 2018; Voynov et al., 2023). However, in some cases, users may not need to generate an entirely new image. Instead, they might be satisfied with making local structural changes to an existing image using partial sketches or simple strokes, e.g., sketch-based image manipulation.

Sketch-based image editing methods can be broadly divided into two categories: mask-based and mask-free approaches. Mask-based methods typically treat the task from an image inpainting perspective, where the user provides not only a mask to define the region for editing, but also several strokes to guide the inpainting process (Yu et al., 2019; Liu et al., 2021; 2024). These strokes, or their features, are often embedded as additional inputs into the network. However, requiring users to manually draw the mask adds extra efforts and may be impractical in certain scenarios. On the other hand, mask-free methods (Zeng et al., 2022) simplify the process by requiring only user-provided strokes for editing, with a mask predictor automatically identifying the region to be modified. Despite the promising results, the aforementioned methods are all based on generative adversarial networks (GANs), which limits their performances. They edit images only from specific domains and often produce artifacts, as shown in the penultimate column of Fig. 1.

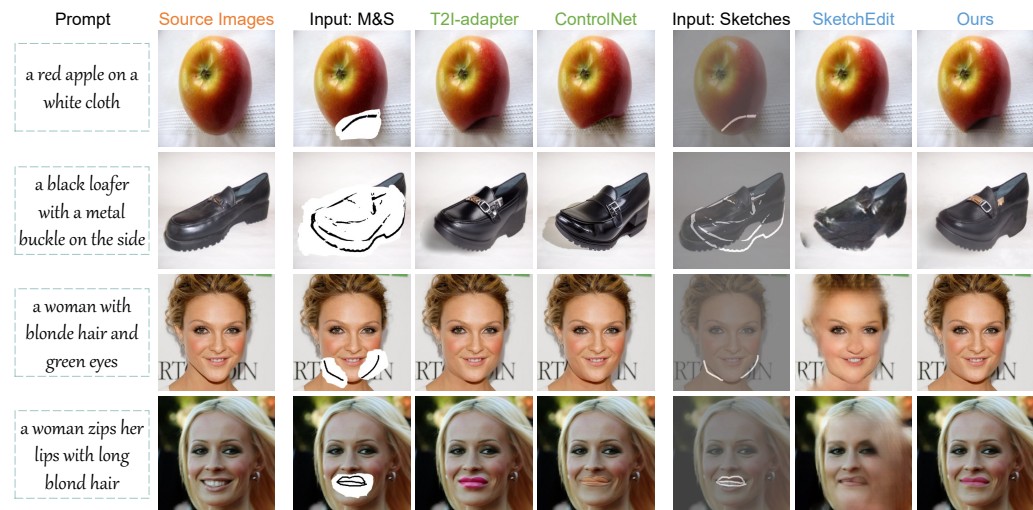

Figure 1: The proposed method enables users to achieve high-quality image manipulation through some strokes without user-provided masks. 'M & S' is short for 'mask and sketch'.

In recent years, diffusion models (Ho et al., 2020; Song et al., 2021b; Rombach et al., 2022) have dominated the field of image generation, achieving the state-of-the-art performance in both image quality and mode coverage. Their powerful generative capabilities have inspired researchers to utilize pre-trained diffusion models for controllable image synthesis. Existing methods have successfully enabled image generation guided by various global conditions (Zhang & Agrawala, 2023; Mou et al., 2024), such as line drawings, semantic maps, and poses. With the involvement of masks and strokes, these conditional control models can modify specific areas of the image to achieve stroke-based editing. However, these methods primarily focus on generating content that aligns with the given conditions without considering consistency with the original image, as the shoes in Fig.1. Furthermore, they require user-provided masks, see Fig.1, which places an additional burden on the user. Therefore, an ideal stroke-based editing technique should simultaneously satisfy the following requirements: 1) The newly generated content needs to align with the stroke while the remaining is consistent with the original image in terms of both content and style. 2) The non-edited regions must remain intact. Due to these factors, stroke-based editing remains a challenging task. Although the DDIM (Song et al., 2021a) technique can preserve the structural information of the original image during editing without a mask, it often leads to significant changes in the image's style and details (Mokady et al., 2023), when classifier-free guidance (CFG) (Ho & Salimans, 2022) is involved.

In this paper, we propose DiffStroke for high-quality, mask-free image editing based on partial sketches. DiffStroke is built upon a conditional control diffusion model, such as ControlNet (Zhang & Agrawala, 2023) and T2I-adapter (Mou et al., 2024), to leverage their strong capability in edge control. We develop a trainable plug-and-play image-stroke fusion (ISF) module and a mask estimation module to address the limitations of previous methods (Zhang & Agrawala, 2023; Mou et al., 2024) in maintaining style consistency and preserving irrelevant areas. As a result, our method ensures that the edited content maintains the same style as the original image, leaves unrelated areas untouched, and achieves high visual quality.

Unlike previous methods (Zhang & Agrawala, 2023; Mou et al., 2024) that only encode the stroke image as input condition embeddings, the proposed ISF module enhances these embeddings by integrating information from the source image using Transformer layers (Vaswani et al., 2017). The stroke and image features are extracted from the sketch adapter (Mou et al., 2024) and the noise predictor of Stable Diffusion (SD) (Rombach et al., 2022), respectively. Leveraging the strong representational capabilities of these pre-trained models, the ISF module achieves effective conditional embeddings without requiring extensive training. With the ISF module, DiffStroke ensures that the newly generated content is structurally aligned with the strokes while maintaining a consistent style with the source image.

To preserve irrelevant areas without requiring user-provided masks, we introduce a mask estimator that automatically determines the regions to be edited based on the image and stroke information. Traditional methods, such as (Zeng et al., 2022), typically train the mask estimator by minimizing the reconstruction error between the target image and the fused result, which is obtained by combining the generated image and the source image using the predicted mask. However, this approach is not suitable for diffusion-based methods, as SD (Rombach et al., 2022) predicts noise in the training stage rather than directly generating the target image. To address this limitation, we leverage Tweedie's formula (Kim & Ye, 2021; Koley et al., 2024a) to estimate a clean latent image during training, which we assume is closer to the target image in the edited regions than the source image. In this way, we can adapt the traditional training method to DiffStroke. Note that the mask estimator is designed to be simple and efficient, requiring only an additional projection layer and a lightweight learnable vector in the shallowest ISF block.

The proposed modules are all plug-and-play, allowing DiffStroke to fully leverage the learned knowledge of the pre-trained conditional control models. Our contributions can be summarized as follows: (i) We propose a mask-free method for high-quality image manipulation with partial sketches. (ii) We develop an image-stroke fusion module to ensure precise control over local shapes while preserving overall style consistency, and an effective training method for mask estimation. (iii) The experimental results on both natural and facial images demonstrate that our method significantly outperforms previous methods.

## 2 RELATED WORK

**Sketch-based visual content generation.** The generation of sketches from images that evoke human abstract concepts is a recurring theme in this field of study. The initial deployment of GANs (Goodfellow et al., 2014) to effect transformations from the domain of real images to that of sketches is a common practice (Isola et al., 2017; Yi et al., 2020; Seo et al., 2023). However, this often necessitates the availability of paired data for training, which can be challenging to collect. Recent work, exemplified by CLIPasso (Vinker et al., 2022), leverages the prior knowledge of pre-trained models (Xing et al., 2023; Vinker et al., 2023), e.g., CLIP (Radford et al., 2021) and SD (Rombach et al., 2022), to facilitate sketch generation at varying degrees of abstraction through the optimization of Bezier curve parameters. However, this approach necessitates a prolonged inference time and disregards the nuances of human drawing style and order in the sketches. Consequently, some studies (Ha & Eck, 2018; Wang et al., 2023; Li et al., 2024) investigate the replication of human drawing habits and the generation of imaginative sketches. The creation of images through the use of sketches has also become a prevalent topic, particularly in conjunction with the advent of diffusion models. In addition to methods based on line drawings (Voynov et al., 2023; Zhang & Agrawala, 2023; Mou et al., 2024), some methods have been investigated about hand-drawn sketches (Koley et al., 2024b) or for target instance editing (Xiao & Fu, 2024). Furthermore, there are also sketch-based video generation tasks, including the synthesis of real video from sketches (Guo et al., 2023) and the animation of sketches (Gal et al., 2024).

**Diffusion model-based image editing.** Along with the recent rapid development of Artificial Intelligence Generated Content (AIGC), numerous image manipulation methods based on diffusion models have emerged (Yang et al., 2023; Huang et al., 2024). One category is the training-based approach, with training subjects that may vary. An example would be the generation of a personalized concept, achieved by optimizing a learnable word embedding (Gal et al., 2023) or fine-tuning the UNet of the diffusion model (Ruiz et al., 2023). Another example is that some additional network layers are trained to achieve style transfer (Ye et al., 2023). Another popular category is the training-free method, which does not require extensive resources. DiffEdit (Couairon et al., 2022) is a straightforward yet productive methodology for approximating the mask of a concept that requires editing for object replacement or removal. This is achieved through the utilization of the attention map that is in alignment with the selected word. Subsequent approaches have also been put forth to achieve image editing by manipulating attention maps (Hertz et al., 2023; Huang et al., 2023). Furthermore, a category of compromises exists that can optimize the Null-text embedding (Mokady et al., 2023) or latent representation (Nam et al., 2024) during the inference process, thereby improving the quality of generation with a little additional time consumption. Note that DiffEdit's method of estimating masks is not suitable for our tasks, because the regions undergoing editing are often localized and difficult to describe in words precisely.

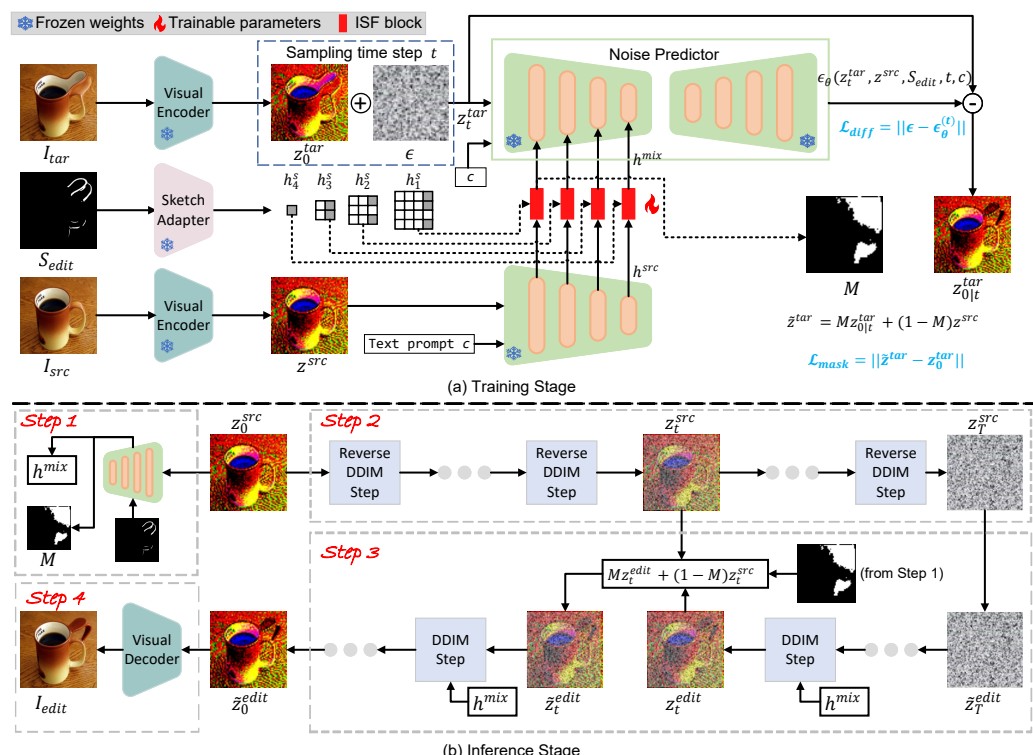

Figure 2: The Overall training pipeline and inference pipeline of DiffStroke. (a) The components of the T2I-adapter (Mou et al., 2024) are frozen and the image-stroke fusing (ISF) blocks are trained. The shallowest of ISFs is also trained for estimating the mask. (b) In the inference phase, the conditional embeddings and the estimated mask are used to generate editing results through the DDIM Inversion (Song et al., 2021a) technique with the inpainting (Lugmayr et al., 2022) method. For the sake of brevity, the ISF blocks are not displayed in Step 1.

## 3 METHODOLOGY

**Overview.** The fundamental objective of DiffStroke is to automatically identify the region to be edited based on the user-supplied image $I_{src}$ and sketch $S_{edit}$, and to generate a conditional embedding to direct the model in the generation of the final editing result $I_{edit}$. The pipeline of the DiffStroke is shown in Fig. 2. The following section presents the particulars of implementing our approach, including the acquisition of paired training data (Section 3.1), the design and training (Section 3.4) of the ISF blocks (Section 3.2 ) and the mask estimator (Section 3.3), and the detailed flow of the inference phase (Section 3.5). To conserve computational resources, DiffStroke is built on the T2I-adapter (Mou et al., 2024) rather than ControlNet (Zhang & Agrawala, 2023).

### 3.1 DATA PREPARATION

Typically, the paired training data of the source image $I_{src}$, the sketch $S_{edit}$, and the editing result $I_{tar}$ are difficult to obtain. Therefore, we adopt a similar strategy to that used in previous methods (Zeng et al., 2022; Xu et al., 2023) to obtain the training data using free-form deformation (FFD) (Sederberg & Parry, 1986), as shown in Fig. 3(a).

Firstly, we initialize the control point grid for FFD. Then, the length and width of the source image $I_{src}$ are normalized, and the control points are distributed uniformly in the range $[0, 1]$ in both the $x$ and $y$ directions. In this context, $g_s$ represents the grid size, e.g., the number of points in a row or a column. Let the grid of control points be $G(i, j)$, where $i, j$ are the grid indices. The initialization expression for the control points is $G(i, j) = \left( \frac{i}{g_s - 1}, \frac{j}{g_s - 1} \right)$. To simulate the free deformation of

Figure 3: (a) The pipeline for obtaining deformed images and conditional sketches for training purposes. (b) Structure of the ISF blocks. The shallowest one is also applied to estimate the mask, i.e., using the path indicated by the dashed arrows.

the image, we randomly shift some of the control points. Let the shifted control point be $\boldsymbol{G}'(i,j)$, which is updated by: $\boldsymbol{G}'(i,j) = \boldsymbol{G}(i,j) + \delta\boldsymbol{d}(i,j)$, where $\delta\boldsymbol{d}(i,j)$ is a random offset vector. We use bi-linear interpolation to implement the deformation. Specifically, given the deformed control point $\boldsymbol{G}'(i,j)$ and the original control point $\boldsymbol{G}(i,j)$, a new pixel coordinate mapping is generated by interpolation. We denote the width and height of the image $I_{src}$ be $W$ and $H$ respectively, and then the coordinate mapping after interpolation in the image is:

$$x'(u,v) = \sum_{i=0}^{g_s-1}\sum_{j=0}^{g_s-1} B_i(u)B_j(v)\boldsymbol{G}'_{\boldsymbol{x}}(i,j), \quad y'(u,v) = \sum_{i=0}^{g_s-1}\sum_{j=0}^{g_s-1} B_i(u)B_j(v)\boldsymbol{G}'_{\boldsymbol{y}}(i,j), \quad (1)$$

where $B_i(u)$ and $B_j(v)$ are the basis functions for bi-linear interpolation, $(x',y')$ represents the new coordinate of each pixel, $(u,v)$ are the normalized coordinates of the source image, and $\boldsymbol{G}'_{\boldsymbol{x}}(i,j)$ and $\boldsymbol{G}'_{\boldsymbol{y}}(i,j)$ are the coordinates of the control point after changes in the $x$ and $y$ directions, respectively. At last, the 'grid_sample' function in PyTorch (Paszke et al., 2019) is employed to implement the new coordinate mapping on the original image, thereby generating the deformed image $\boldsymbol{I}_{tar}$. Please refer to our submitted code for more details.

To get the conditional sketch $\boldsymbol{S}_{edit}$, we initially calculate the moved distance of the control points:

$$\Delta\boldsymbol{G}(x,y) = ||\boldsymbol{G}(x,y) - \boldsymbol{G}'(x,y)||. \quad (2)$$

Subsequently, the deformation field $\Delta\boldsymbol{G}$ is extended to the resolution of the entire image by bi-linear interpolation to get $\Delta\hat{\boldsymbol{G}}(x,y)$, thereby generating a mask $\hat{M}$:

$$\hat{M}(x,y) = \begin{cases} 1 & \text{if } \Delta\hat{\boldsymbol{G}}(x,y) > 0.05, \\ 0 & \text{otherwise.} \end{cases} \quad (3)$$

The mask $\hat{M}$ determines whether each pixel location is in a deformation region or not. We then leverage PidiNet (Su et al., 2021) to extract edge map $\boldsymbol{S}_{src}$ and $\boldsymbol{S}_{tar}$ from $\boldsymbol{I}_{src}$ and $\boldsymbol{I}_{tar}$, respectively. Finally, the conditional sketch $\boldsymbol{S}_{edit}$ is obtained by $\hat{M} \odot (\boldsymbol{S}_{tar} - \boldsymbol{S}_{src})$.

## 3.2 Aggregating the Image and Sketch Information

In this study, we employ the sketch-controlled diffusion model T2I-adapter (Mou et al., 2024) as the base, for the sketch-based image editing task. In the generation of image $\boldsymbol{I}_{tar}$ (or $\boldsymbol{I}_{edit}$), the T2I-adapter extracts the features $\boldsymbol{h}^s = [\boldsymbol{h}_1^s, \boldsymbol{h}_2^s, \boldsymbol{h}_3^s, \boldsymbol{h}_4^s]$ from the sketch $\boldsymbol{S}_{tar}$ at four distinct layers. These are then summed with the hidden layer features $\boldsymbol{h}_{(t)}^{tar} = [\boldsymbol{h}_{1(t)}^{tar}, \boldsymbol{h}_{2(t)}^{tar}, \boldsymbol{h}_{3(t)}^{tar}, \boldsymbol{h}_{4(t)}^{tar}]$ of the noise predictor $\epsilon_\theta$. The embeddings $\boldsymbol{h}^s$ serve to guide the generation process at the time step $t$. To mitigate the potential loss of stylistic content resulting from the exclusive utilization of sketches as conditioning variables, we augment the conditional control embeddings and introduce the ISF block. The structure of the ISF block is illustrated in Fig. 3(b). Given the powerful representations afforded by SD's UNet, we leverage this model to extract the features $\boldsymbol{h}^{src} = [\boldsymbol{h}_1^{src}, \boldsymbol{h}_2^{src}, \boldsymbol{h}_3^{src}, \boldsymbol{h}_4^{src}]$ of the

latent source image $z^{src}$ from the same layers as $h_{(t)}^{tar}$, thereby capturing the style content conditions. Subsequently, the source image embedding $h_i^{src}$ and the sketch feature $h_i^s$ are added and fed into three transformer layers (Vaswani et al., 2017). This enables the interaction within different tokens through the self-attention mechanism and the feed-forward networks. The transformer block then output the control embedding, $h^{mix} = [h_1^{mix}, h_2^{mix}, h_3^{mix}, h_4^{mix}]$. Ultimately, instead of the sketch embeddings $h^s$, the augmented one $h^{mix}$ will be employed for model training and image generation through a summation with $h_{(t)}^{tar}$.

### 3.3 ESTIMATING THE EDITING REGIONS

To equip the DiffStroke with the functionality of an estimation mask, additional designs are created for the first ISF block. The selection of this particular ISF block is based on two considerations. Firstly, shallow features reflect more specific local details rather than global semantics. Secondly, the height and width of the feature $h_1^{src}$ are consistent with the latent source image $z^{src}$. The method is implemented by introducing a learnable vector $v^* \in \mathbb{R}^{(64 \times 64) \times 16}$ as additional channels for $h_1^{src} \in \mathbb{R}^{64 \times 64 \times 320}$. We utilize the information interaction capabilities of the ISF block to enable $v^*$ to recognize the specific editing regions. A multi-layer perceptron (MLP), followed by transformer layers, produces the final output mask $M \in \mathbb{R}^{64 \times 64}$, as illustrated in Fig. 2(a). The process can be formalized as:

$$\bar{h}_1^{src} = f_{con}(f_{rs}(h_1^{src}), v^*), \quad (\bar{h}_1^{mix}, \bar{v}^*) = f_{ISF}(\bar{h}_1^{src}),$$
$$M = f_{MLP}(f_{rs}(\bar{v}^*)), \quad h_1^{mix} = f_{rs}(\bar{h}_1^{mix}), \tag{4}$$
$$\bar{h}_1^{src} \in \mathbb{R}^{(64 \times 64) \times 336}, \quad \bar{h}_1^{mix} \in \mathbb{R}^{(64 \times 64) \times 320}, \quad \bar{v}^* \in \mathbb{R}^{(64 \times 64) \times 16},$$

where $f_{con}(\cdot)$ and $f_{rs}(\cdot)$ respectively denote the vectors concatenated process and the reshape operation, $f_{ISF}(\cdot)$ is the first ISF block, and $f_{MLP}(\cdot)$ represents the MLP for producing mask $M$. To reduce the number of parameters, we apply a one-layer convolutional neural network with a kernel size of $3 \times 3$ and a stride of $1$ to implement $f_{MLP}(\cdot)$.

### 3.4 TRAINING THE COMPONENTS OF DIFFSTROKE

In the training phase, the parameters of the ISF blocks including the mask prediction network are optimized. We first encode the source image $I_{src}$ and the deformed image $I_{tar}$ into the latent representations $z^{src}$ and $z^{tar}$ (i.e., $z_0^{tar}$), while leveraging the sketch adapter to get the features $h^s = \mathcal{A}(S_{edit})$. The vectors $z^{src}$ and $h^s$ are used to calculate the conditional embeddings $h^{mix}$. In each training step, the noise $\epsilon \sim \mathcal{N}(0, I)$ and the time step $t$ are randomly sampled to introduce noise into $z^{tar}$:

$$z_t^{tar} = \sqrt{\bar{\alpha}_t} z_0^{tar} + \sqrt{1 - \bar{\alpha}_t} \epsilon, \tag{5}$$

where $\bar{\alpha}_t$ denotes the compound of the noise schedule $\alpha_t$. DiffStroke injects the conditional embeddings $h^{mix}$ to noise predictor and adopts the same strategy as commonly used conditional control networks (Zhang & Agrawala, 2023; Mou et al., 2024) to train the ISF blocks:

$$\mathcal{L}_{diff} = ||\epsilon - \epsilon_\theta(z_t^{tar}, z^{src}, S_{edit}, t, c)||_2^2, \tag{6}$$

where $c$ denotes the text prompt. To train the mask estimator, the Tweedie's formula (Kim & Ye, 2021; Koley et al., 2024a) is initially employed:

$$z_{0|t}^{tar} = \frac{z_t^{tar} - \sqrt{1 - \bar{\alpha}_t} \epsilon_\theta(z_t^{tar}, z^{src}, S_{edit}, t, c)}{\sqrt{\bar{\alpha}_t}}. \tag{7}$$

This yielded the requisite estimated clean latent image $z_{0|t}^{tar}$. Subsequently, $z_{0|t}^{tar}$ and $z^{src}$ are combined to get the output $\tilde{z}^{tar} = M z_{0|t}^{tar} + (1 - M)(\hat{M} z^{src} + (1 - \hat{M}) z^{tar})$. The mask generated by Eq. 3 is employed to circumvent the confounding influence of the deformation in the irrelevant region induced by the FFD on the training process. The mask estimator can be optimized by minimizing the errors between $z_0^{tar}$ and $\tilde{z}^{tar}$:

$$\mathcal{L}_{mask} = ||\tilde{z}^{tar} - z_0^{tar}||_2^2. \tag{8}$$

To strengthen the control of the edge conditions, we introduce an additional regular term:

$$\mathcal{L}_{edge} = ||h^{mix} - \mathcal{A}(S_{tar})||_2^2, \tag{9}$$

where $\boldsymbol{S}_{tar}$ is the edge map of the target image $\boldsymbol{I}_{tar}$. The overall loss function of DiffStroke is:

$$\mathcal{L} = \mathcal{L}_{diff} + 2.5\mathcal{L}_{mask} + 0.25\mathcal{L}_{edge}. \tag{10}$$

In practical, we add noise to $\boldsymbol{z}^{src}$ ($t = 273$) when extracting the image features, which more accurately reflect the edge features, as recommended by existing literature (Koley et al., 2024a).

### 3.5 Editing images by DiffStroke

In the inference stage, users provide the source image $\boldsymbol{I}_{src}$ and the stroke $\boldsymbol{S}_{edit}$ that are encoded to the latent source image $\boldsymbol{z}_0^{src}$ and the embedding $h^{src}$. Subsequently, DiffStroke employs the DDIM reverse step (Song et al., 2021a) to generate the noise vectors $\boldsymbol{z}_0^{src}, \boldsymbol{z}_1^{src}, ..., \boldsymbol{z}_T^{src}$ for distinct time steps, produces the conditional embeddings $\boldsymbol{h}^{mix}$, and estimates the mask mask $\boldsymbol{M}$. We take $\boldsymbol{z}_T^{src}$ as the initial noise $\tilde{z}_T^{edit}$ for the DDIM denoising process. The process in the time step $t$ is as follows:

$$\boldsymbol{z}_{t-1}^{edit} = \sqrt{\bar{\alpha}_{t-1}}(\frac{\tilde{\boldsymbol{z}}_t^{edit} - \sqrt{1 - \bar{\alpha}_t}\epsilon_\theta^{(t)}}{\bar{\alpha}_t}) + \sqrt{1 - \bar{\alpha}_{t-1}}\epsilon_\theta^{(t)},$$
$$\tilde{\boldsymbol{z}}_{t-1}^{edit} = \boldsymbol{M}\boldsymbol{z}_{t-1}^{edit} + (1 - \boldsymbol{M})\boldsymbol{z}_{t-1}^{src}, \tag{11}$$

where $\epsilon_\theta^{(t)}$ denotes $\epsilon_\theta(\tilde{\boldsymbol{z}}_t^{edit}, \boldsymbol{z}^{src}, \boldsymbol{S}_{edit}, t, c)$. Ultimately, the latent image $\tilde{z}_0^{edit}$ is obtained and subsequently decoded to generate the edited image $I_{edit}$ as depicted in Fig. 2(b). To maintain the integrity of the unedited regions, the mask $\boldsymbol{M}$ is up-sampled and employed to fuse $\boldsymbol{I}_{edit}$ and $\boldsymbol{I}_{src}$.

## 4 Experiments

**Datasets.** We test model performance on natural and facial image datasets like previous sketch-based image manipulation methods (Liu et al., 2021; Zeng et al., 2022). For training on generic scenes, we opted for the smaller Sketchy dataset (11,250 images for training) (Sangkloy et al., 2016) due to its ease of training, rather than the larger Places2 dataset (1.8 million images) (Zhou et al., 2017). However, to ensure fairness compared to methods trained on Places2, we conducted quantitative experiments using 2,000 randomly selected images from the Places2 validation set. For facial image manipulation, we used the CelebA-HQ dataset (Karras, 2017), training on 28,000 images and testing on other 2,000 images. To better capture face deformation, we followed a strategy similar to SketchEdit (Zeng et al., 2022), replacing grid control points with face landmarks detected via 'dlib' in 80% of the training cases. We swapped source and target images with a 50% probability, adjusting the conditional sketches accordingly. For quantitative analysis, we adhered to the SketchEdit scheme: 1) Deforming the source image $\boldsymbol{I}_{src}$ to obtain the deformed image $\boldsymbol{I}_{def}$, producing a sketch $\boldsymbol{S}_{def}$ of the deformed region. 2) Each model generates a new image $\boldsymbol{I}_{edit}$ conditional on $\boldsymbol{I}_{def}$ and $\boldsymbol{S}_{def}$. 3) Calculating metrics between the ground truth $\boldsymbol{I}_{src}$ and the model output $\boldsymbol{I}_{edit}$. We leverage blip2 (Li et al., 2023) to generate captions corresponding to the images automatically.

**Implementation details.** The dimension of the Transformers' feed-forward network in DiffStroke's ISF blocks is $1024$. We trained DiffStroke using the AdamW optimizer (Loshchilov et al., 2017) with $\beta_1 = 0.9$ and $\beta_2 = 0.999$. The learning rate was set to $0.0001$ and the batch size is $4$. A total of 170,000 steps were trained on the natural image dataset, which was then used to train an additional 30,000 steps on the CelebA-HQ dataset for face manipulation. The version of SD (Rombach et al., 2022) is v1.5. We set the DDIM step (Song et al., 2021a) to 50 by default. All experiments are conducted on a single Nvidia A100 40G.

**Competitors.** In addition to the mask-free SketchEdit (Zeng et al., 2022), we also conducted a comparative analysis of the state-of-the-art models that require mask participation. These include GAN-based DeepFill-v2 (Yu et al., 2019), DeFLOCNet (Liu et al., 2021), and SketchRefiner (Liu et al., 2024), as well as diffusion model-based ControlNet (Zhang & Agrawala, 2023) and T2I-adapter (Mou et al., 2024). Two approaches are employed to provide masks for these methods: computation using Eq. 3 and estimation via DiffStroke (followed by *, e.g., SketchRefiner*).

### 4.1 Qualitative Analysis

Fig. 4 presents the manipulation results of natural images. The proposed DiffStroke model exhibits favorable outcomes for both shape control and style retention. DiffStroke, T2I-adapter (Mou et al.,

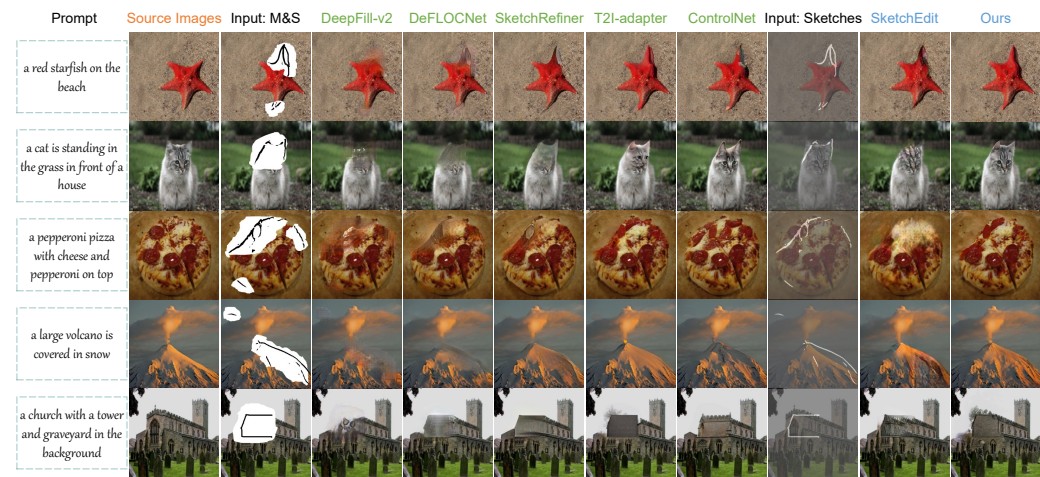

Figure 4: Examples of edits on natural images. Our method and SketchEdit (Zeng et al., 2022) are not required for user-provided masks. 'M & S' is short for 'mask and sketch'.

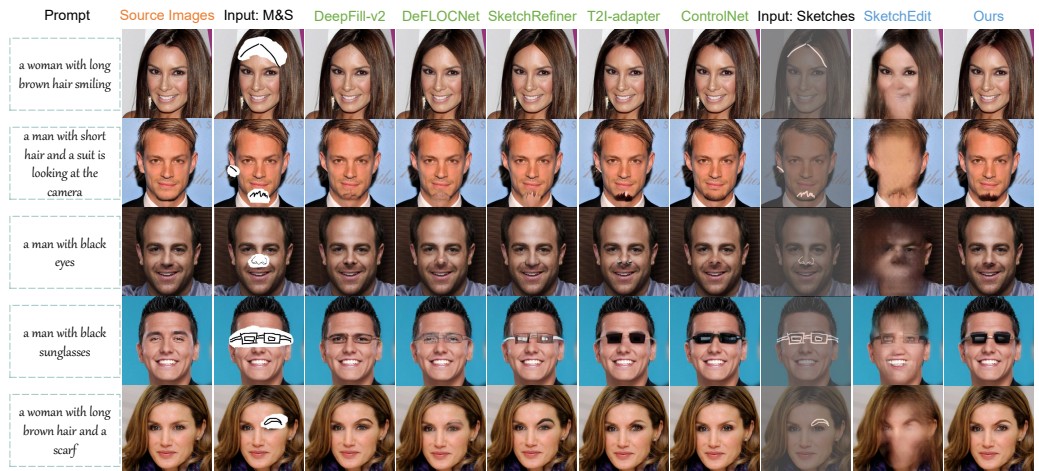

Figure 5: Examples of edits on facial images. Our method and SketchEdit (Zeng et al., 2022) are not required for user-provided masks. 'M & S' is short for 'mask and sketch'.

2024), and ControlNet (Zhang & Agrawala, 2023) are capable of producing more high-quality than GANs. This is attributable to the potent generative capabilities inherent in SD (Rombach et al., 2022). Although SketchRefiner (Liu et al., 2024) produces superior results to other GAN-based methods, its performance is still inadequate, producing artifacts, when confronted with complex scenes such as 'cat's head'. The advantage of DiffStroke over T2I-adapter and ControlNet, in addition to being mask-free, is in the effectiveness of ISF blocks for feature fusion to enhance control embedding. To illustrate, the edited result of the T2I-adapter contains two cat mouths (the second row of Fig. 4), the wall added to the church is too dark in color, and there is no connection between the edited and non-edited areas (the latest row of Fig. 4). Furthermore, ControlNet is not effective in modifying the cat's ears or the shape of the pizza.

For face manipulation, the discrepancy between GANs and diffusion models is decreasing, as shown in Fig. 5. When executing simple editing operations, e.g., modifying a hairstyle (the first row of Fig. 5), the majority of techniques demonstrate remarkable efficacy. Conversely, for more intricate tasks, such as adding a beard (the second row of Fig. 5) or wearing eyeglasses (the fourth row of Fig. 5), only our method is capable of striking a satisfactory balance between the quality of the generated

| Method | Mask | Plces2 | | | | CelebA-HQ | | | |
|---|---|---|---|---|---|---|---|---|---|
| | | FID (↓) | PSNR (↑) | SSIM (↑) | LPIPS (↓)) | FID (↓) | PSNR (↑) | SSIM (↑) | LPIPS (↓) |
| $I_{def}$ | - | 6.51 | 29.14 | 0.9192 | 0.0383 | 3.21 | 29.60 | 0.9448 | 0.0201 |
| DeepFill-v2 | ✓ | 10.42 | 27.82 | 0.9065 | 0.0806 | 6.37 | 30.00 | 0.9334 | 0.0441 |
| DeepFill-v2* | ✓ | 8.50 | 29.91 | 0.9257 | 0.0704 | 5.45 | 30.97 | 0.9452 | 0.0362 |
| DeFLOCNet | ✓ | 8.72 | 27.67 | 0.9073 | 0.0739 | 5.45 | 30.37 | 0.9381 | 0.0345 |
| DeFLOCNet* | ✓ | 6.25 | 29.99 | **0.9290** | 0.0652 | 4.53 | **32.71** | **0.9624** | 0.0281 |
| SketchRefiner | ✓ | 5.36 | 29.51 | 0.9220 | 0.0361 | 2.95 | 30.35 | 0.9437 | 0.0253 |
| SketchRefiner* | ✓ | 4.88 | 30.05 | 0.9249 | 0.0311 | 2.16 | 31.46 | 0.9547 | 0.0188 |
| ControlNet | ✓ | 5.39 | 27.94 | 0.9165 | 0.0417 | 3.25 | 29.56 | 0.9406 | 0.0256 |
| ControlNet* | ✓ | 5.35 | 29.11 | 0.9208 | 0.0384 | 3.07 | 30.52 | 0.9507 | 0.0214 |
| T2I-adapter | ✓ | 6.88 | 28.58 | 0.9202 | 0.0437 | 4.01 | 30.21 | 0.9495 | 0.0237 |
| T2I-adapter* | ✓ | 5.30 | 29.54 | 0.9240 | 0.0327 | 3.03 | 30.79 | 0.9547 | 0.0200 |
| SketchEdit | ✗ | 6.27 | 29.28 | 0.9148 | 0.0437 | 45.36 | 19.09 | 0.6741 | 0.2734 |
| SketchEdit* | ✓ | 5.75 | 29.73 | 0.9222 | 0.0407 | 16.55 | 28.54 | 0.9421 | 0.0378 |
| DiffStroke (ours) | ✗ | **4.78** | **30.09** | 0.9256 | **0.0304** | **1.99** | 32.04 | 0.9571 | **0.0156** |

Table 1: Quantitative comparison on synthetic samples from CelebA-HQ (Karras, 2017) and Places2 validation sets (Zhou et al., 2017). The image resolution used to calculate the metrics is $256 \times 256$. The first line of results is the discrepancy between the deformed images and the source images.

| Method | Mask | Plces2 | | | | CelebA-HQ | | | |
|---|---|---|---|---|---|---|---|---|---|
| | | FID (↓) | PSNR (↑) | SSIM (↑) | LPIPS (↓)) | FID (↓) | PSNR (↑) | SSIM (↑) | LPIPS (↓) |
| $I_{def}$ | - | 5.54 | 28.96 | 0.9165 | 0.0409 | 2.52 | 29.33 | 0.9468 | 0.0278 |
| ControlNet | ✓ | 5.41 | 28.50 | 0.9235 | 0.0508 | 3.71 | 30.00 | 0.9486 | 0.0379 |
| ControlNet* | ✓ | 5.40 | 29.81 | 0.9281 | 0.0471 | 3.60 | 30.99 | 0.9560 | 0.0305 |
| T2I-adapter | ✓ | 5.83 | 29.39 | 0.9297 | 0.0459 | 3.92 | 30.73 | 0.9569 | 0.0312 |
| T2I-adapter* | ✓ | 5.32 | 30.26 | 0.9313 | 0.0408 | 3.46 | 31.25 | 0.9597 | 0.0274 |
| DiffStroke (ours) | ✗ | **4.80** | **30.86** | **0.9330** | 0.0392 | 2.24 | **32.56** | **0.9623** | **0.0238** |

Table 2: Quantitative comparison on synthetic samples from CelebA-HQ (Karras, 2017) and Places2 validation sets (Zhou et al., 2017). The image resolution used to calculate the metrics is $512 \times 512$.

output and control conditions provided by the users. We observed that SketchEdit produces lots of artifacts in irrelevant regions. This can be attributed to inaccurate mask predictions and insufficient generation capabilities. More editing results produced by DiffStroke are provided in Appendix D.

## 4.2 QUANTITATIVE ANALYSIS

As the GAN-based methods utilize an image resolution of 256x256, while the diffusion models have a resolution of 512x512, we present the metrics of the metrics at both resolutions, as illustrated in Tables 1 and Table 2. We deflate the image by bi-linear interpolation. The weight of CFG (Ho & Salimans, 2022) for diffusion models is set to 3.0 which is a compromise between generation quality and style consistency. Overall, DiffStroke exhibits superior performance compared to the other methods in terms of the natural scene and face datasets. We also find that mask-required methods with masks estimated using DiffStroke (method names ending in '*') demonstrate superior performance compared to masks generated by Eq. 3. This observation implies that, through training, the mask estimator is capable of accurately identifying the regions that require editing, rather than merely fitting the masks produced by Eq. 3. Meanwhile, SketchEdit (Zeng et al., 2022) can obtain better metrics with the estimated masks by DiffStroke instead of their predictions. This implies the superiority of our mask estimator.

Although DeFLOCNet (Liu et al., 2021) shows marginally higher PSNR and SSIM values than DiffStroke at the resolution of 256x256, FID (Heusel et al., 2017) and LPIPS (Zhang et al., 2018) exhibit a notable weakness compared to DiffStroke which indicates our method still significantly outperforms DeFLOCNet. Among all the methods, one GAN-based model that is metrically similar to ours and outperforms other diffusion models is SketchRefiner (Liu et al., 2024). This is because SketchRefiner has been trained specifically on these two datasets, whereas ControlNet (Zhang & Agrawala, 2023) with T2I-adapter (Mou et al., 2024) represents a relatively more general approach. Furthermore, the quantitative experiments are conducted at a relatively small deformation scale to ensure the realism of the deformed images $I_{def}$ (as shown in Appendix C), resulting in smaller regions that need to be edited. This allows SketchRefiner to perform the task effectively. It is noteworthy that SketchEdit displays considerably inferior performance on the CelebA-HQ dataset

| Method | Plces2 | | | | CelebA-HQ | | | |
|---|---|---|---|---|---|---|---|---|
| | FID (↓) | PSNR (↑) | SSIM (↑) | LPIPS (↓) | FID (↓) | PSNR (↑) | SSIM (↑) | LPIPS (↓) |
| w/o ISF & Mask | 38.65 | 18.78 | 0.6545 | 0.3977 | 44.28 | 20.55 | 0.7617 | 0.2736 |
| w/o Mask | 30.86 | 21.61 | 0.7550 | 0.2799 | 29.22 | 23.53 | 0.8272 | 0.1863 |
| w/o ISF | 5.32 | 30.26 | 0.9313 | 0.0408 | 3.46 | 31.25 | 0.9597 | 0.0274 |
| Ours-full | **4.80** | **30.86** | **0.9330** | **0.0392** | **2.24** | **32.56** | **0.9623** | **0.0238** |

Table 3: Ablation on design.

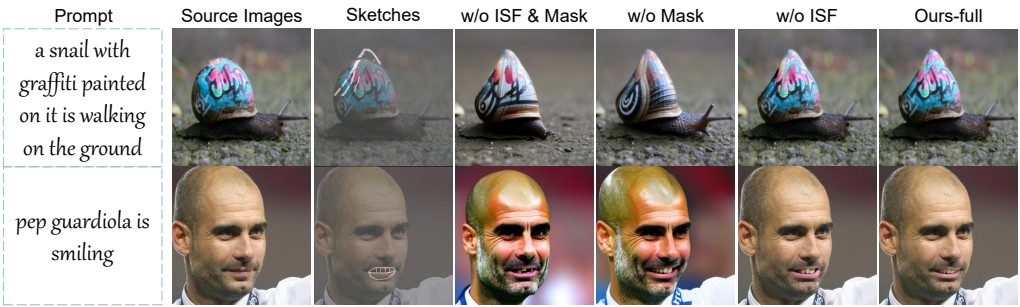

Figure 6: Visual ablation of different settings.

in comparison to the Places2 dataset. This may be attributed to the sensitivity of SketchEdit to facial images, which is discussed in Appendix A.

## 4.3 ABLATION STUDY

This section will examine the role of ISF blocks to enhance stroke embeddings and the advantages of using masks over the pure DDIM Inversion (Song et al., 2021a). **1) Importance of mask estimator.** The numerical results in Table 3 show that the utilization of masks exerts a considerable influence on the metrics. Despite the generation of high-quality images through the CFG (Ho & Salimans, 2022) in the absence of masks, as illustrated in Fig. 6, the style and structure have resulted in notable discrepancies. Qualitative and quantitative results demonstrate the difficulty of ensuring the invariance of non-edited regions by relying only on DDIM Inversion (Song et al., 2021a). **2) The effectiveness of ISF blocks.** As shown in Table 3, the introduction of ISF blocks has been demonstrated to markedly enhance the pertinent quantitative metrics, particularly for the face dataset. From a visual performance perspective, the ISF blocks preserve the structure of the source image effectively in the absence of mask cooperation. When masks are involved, in addition to maintaining style, the ISF blocks facilitate the generation of higher-quality local content, such as Pep's neat teeth. In addition to these, we also provide a discussion on the regular term, please refer to the appendix B.

## 5 CONCLUSIONS AND LIMITATIONS

The paper investigates the potential of implementing high-quality mask-free image manipulation with partial sketches based on a conditional control diffusion model. We propose a plug-and-play model named DiffStroke. To achieve the preservation of style and the creation of controllable structures for the editing results, we introduce the ISF module for the fusion of image-sketch information and a training method for the estimation of masks. Both qualitative and quantitative results demonstrate the effectiveness of our approach. We also provide further experimental results and analyses in the Appendix, which readers may find beneficial in gaining more insight. Meanwhile, there are still some limitations to our approach that warrant further exploration. One challenge is guiding the model to generate results that align with human expectations based on strokes, rather than merely producing textures that fit the sketch structure in some cases. Another challenge is mask-free object replacement by text and strokes. This task requires a more powerful model capacity to achieve more flexible controllable editing such as replacing a specific bush in a garden with a wooden fence.

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

APPENDIX

## A  THE SENSITIVITY OF SKETCHEDIT TO FACIAL IMAGES

SketchEdit (Zeng et al., 2022) for face manipulation is trained and tested on the CelebA-HQ (Karras, 2017) dataset with 256x256 pixels. The default size of the images in CelebA-HQ is 1024x1024, which means we need to down-sample it using interpolation. In this paper, we use the official test code and pre-trained weights provided by SketchEdit to evaluate its performance. When we edit with the facial images provided by the official SketchEdit GitHub repository, we can produce clear results. However, when we find the same image from the CelebA-HQ dataset as provided in the official demo and downsize it, it fails to produce results of similar quality. As illustrated in Fig. 7, a variety of interpolation techniques were employed, including nearest neighbor, bi-linear, area, bi-cubic, and Lanczos interpolation. However, these approaches yielded only blurred results.

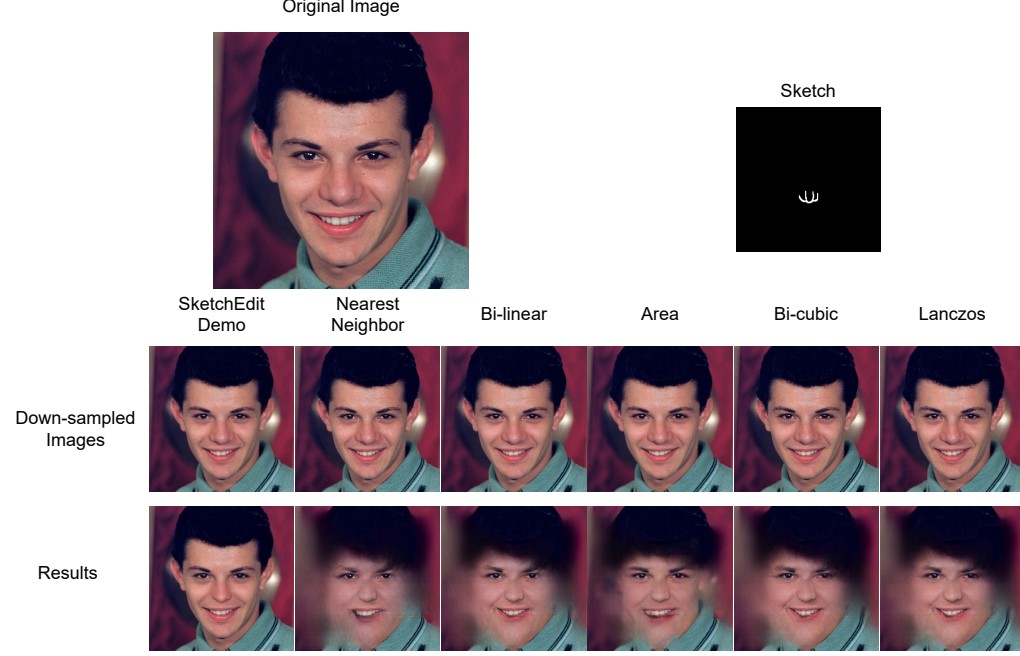

Figure 7: The results of facial image manipulation of SketchEdit(Zeng et al., 2022). Various methods are used to downsample an image from CelebA-HQ (Karras, 2017) with 1024x1024 pixels to 256×256 pixels. 'Original Image' denotes the image from the CelebA-HQ dataset with a resolution of 1024×1024. The 'Sketch' image and the 'SketchEdit Demo' image are from the official SketchEdit GitHub repository.

Unfortunately, only four facial images ('.png') are provided in their official open-source repository, which is not enough for quantitative testing. Although the authors of SketchEdit have indicated in the 'README.md' file of their official repository, which was updated on 1 June 2022, that training data and training-related code will be made available, this has not been done to date. Without the extensive training data processing specific code and training details mentioned in their paper, it is difficult to retrain it according to our deflation method. A similar situation where SketchEdit has a large gap between the face dataset and the natural image dataset on the metrics is also present in SketchRefiner's paper (Liu et al., 2024).

## B  EFFECTIVENESS OF THE REGULAR TERM

In this section, we discuss the impact of the regular term $\mathcal{L}_{edge}$ in eq. 9. Table 4 reports the metrics obtained with and without the use of the regular term. It has been demonstrated that the metrics

exhibit slightly superior performance when the embedding $h^{mix}$ is not constrained, as opposed to introducing the regular term $\mathcal{L}_{edge}$ during the training process. This is because the model may be capable of focusing more on the color and texture information of the conditional image, thereby guiding the generated results to a greater extent in maintaining the style. However, this can result in a loss of edge control, as shown in Fig. 8. In the event of $\mathcal{L}_{edge}$ non-participation in the training, the edited result may not accurately reflect the intended deformation, as illustrated by the spoon in Fig. 8. Furthermore, additional content may emerge in the edited region that is not strictly aligned with the sketch, such as the feathers at the swan's tail and the lines at the banana stalk.

| Method | Plces2 | | | | CelebA-HQ | | | |
|---|---|---|---|---|---|---|---|---|
| | FID ($\downarrow$) | PSNR ($\uparrow$) | SSIM ($\uparrow$) | LPIPS ($\downarrow$) | FID ($\downarrow$) | PSNR ($\uparrow$) | SSIM ($\uparrow$) | LPIPS ($\downarrow$) |
| w/o $\mathcal{L}_{edge}$ | 4.77 | 30.78 | 0.9326 | 0.0395 | 1.98 | 32.69 | 0.9622 | 0.0237 |
| w $\mathcal{L}_{edge}$ | 4.80 | 30.86 | 0.9330 | 0.0392 | 2.24 | 32.56 | 0.9623 | 0.0238 |

Table 4: Quantitative results on the effective of $\mathcal{L}_{edge}$.

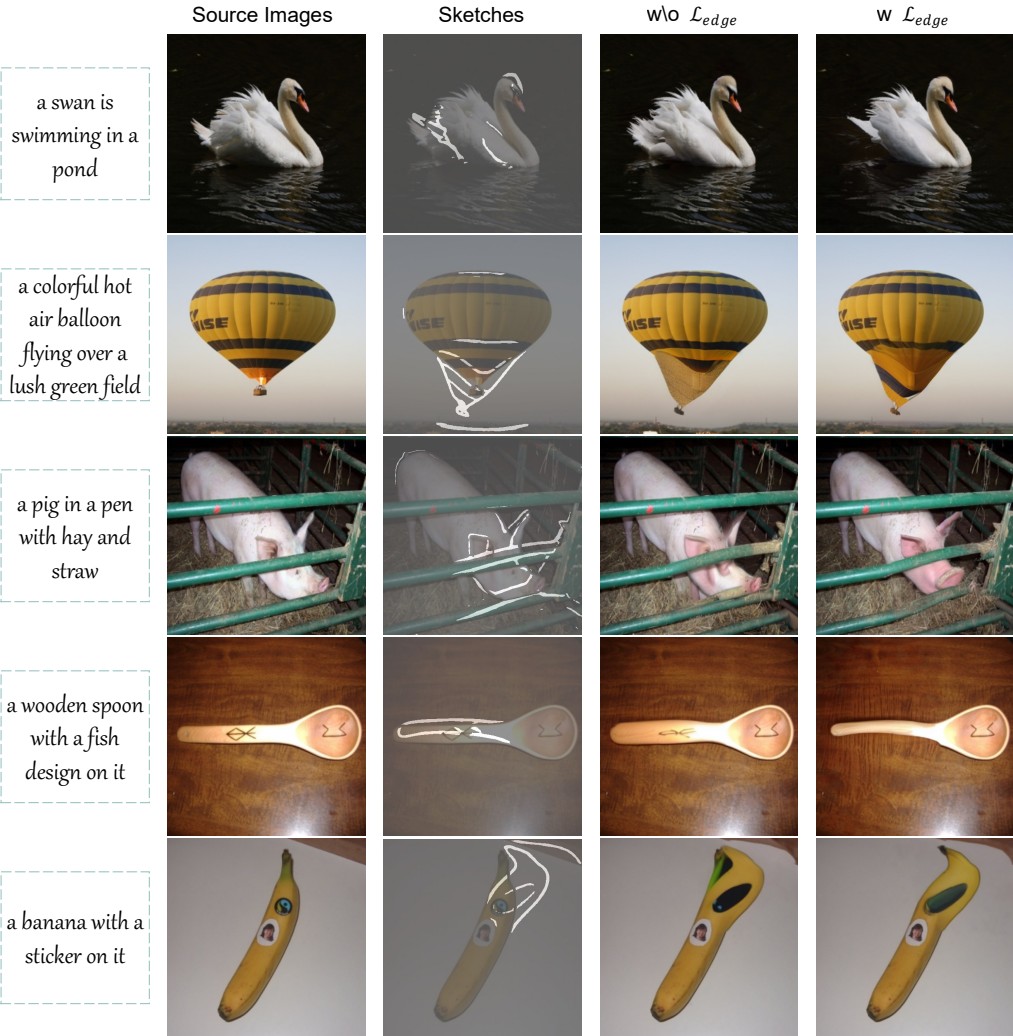

Figure 8: Example of the edited images by the models trained with/without $\mathcal{L}_{edge}$.

## C VISUALIZATION OF THE EDITED IMAGES IN QUANTITATIVE ANALYSIS

To provide a more illustrative representation of the recreated results obtained from the deformed images in the quantitative analysis, we present some examples from the Places2 (Zhou et al., 2017) and CelebA-HQ (Karras, 2017) datasets in Fig. 9 and 10, respectively. Meanwhile, we provide the masks estimated by DiffStroke during the editing process.

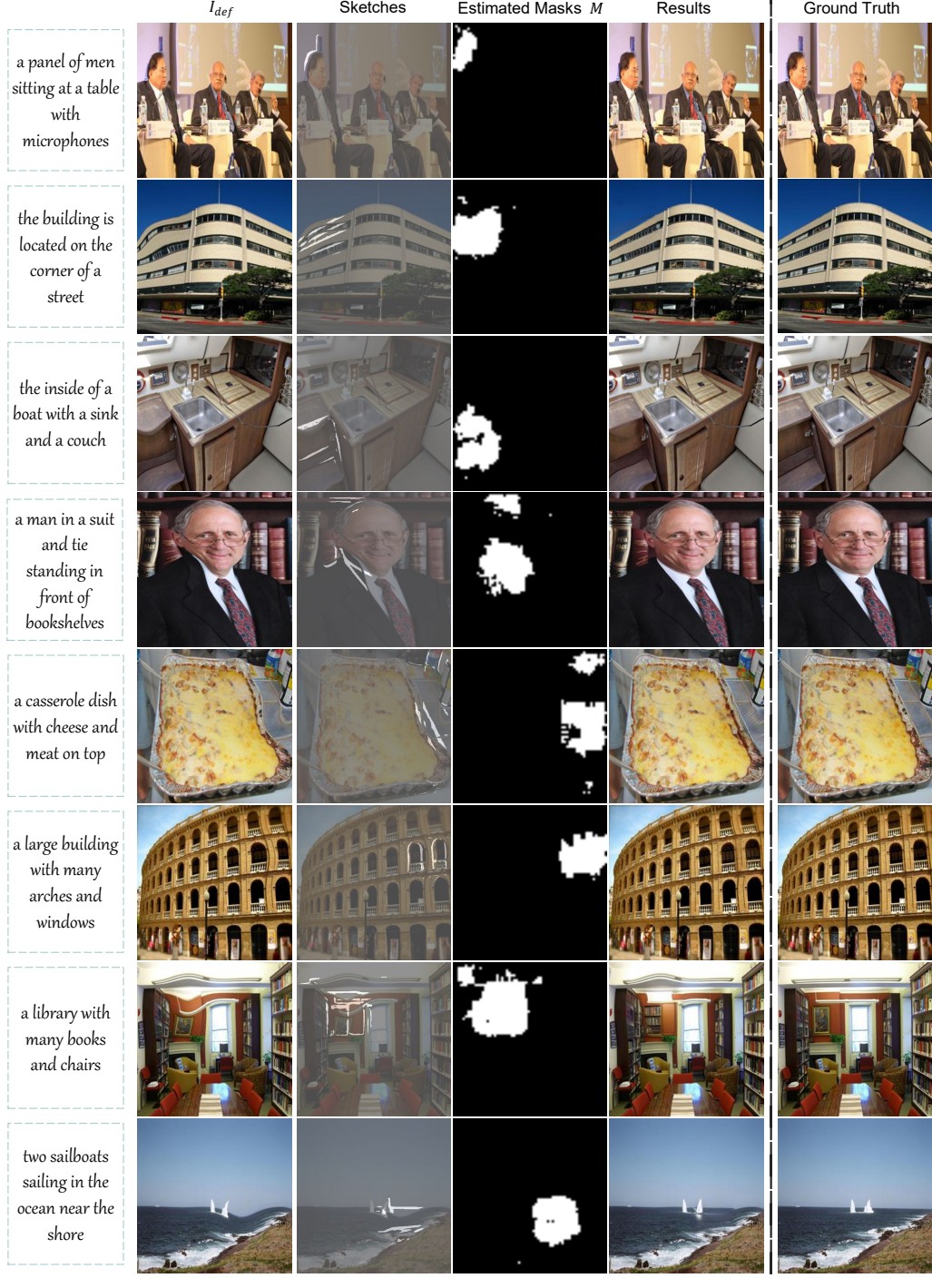

Figure 9: Recreate natural images from deformed images. Masks are estimated by DiffStroke.

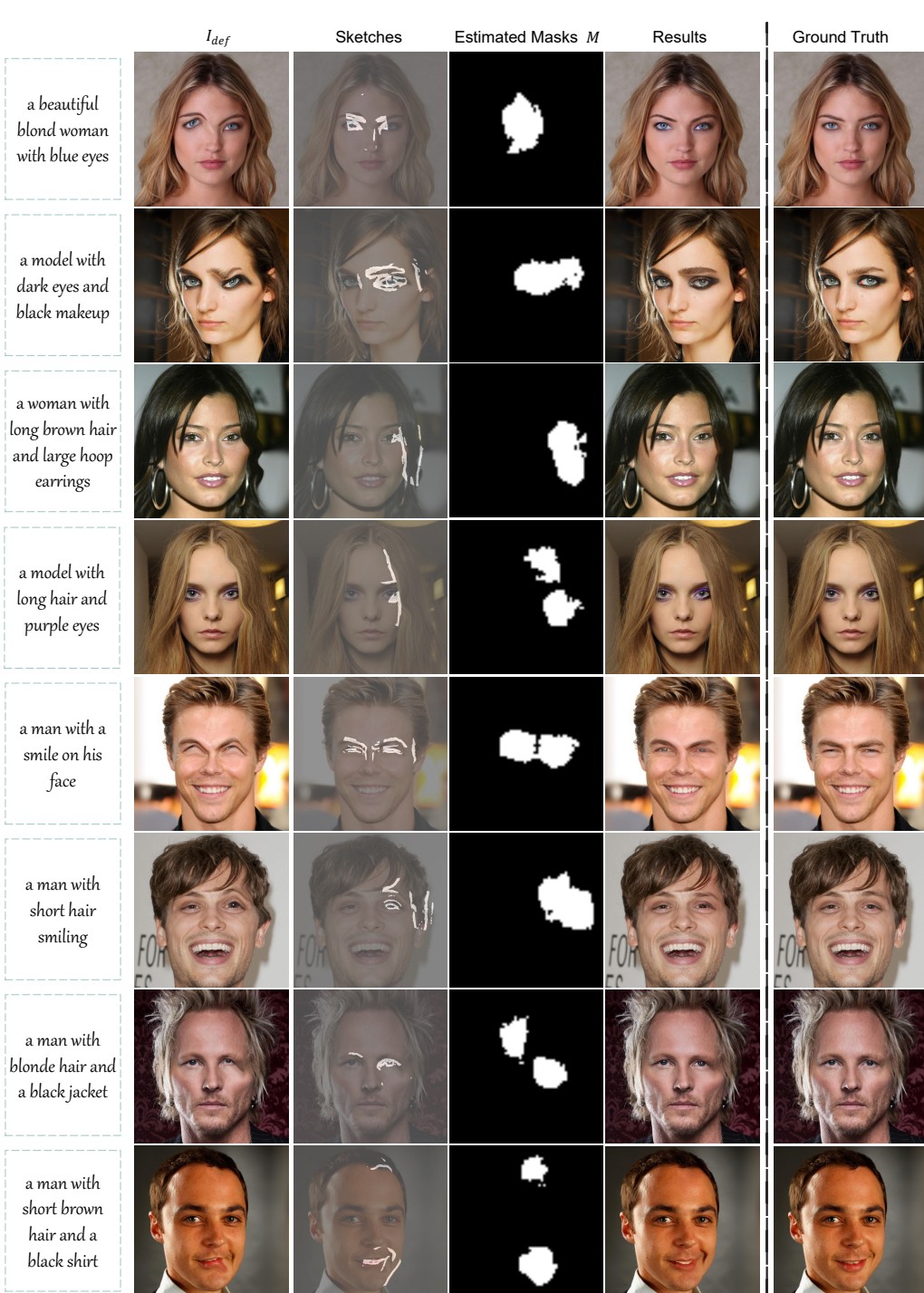

Figure 10: Recreate facial images from deformed images with sketches by our model. Masks are estimated by DiffStroke.

Limited by the file size that can be uploaded, we are currently only able to provide the code of DiffStroke. The data pertinent to the quantitative experiments, including the deformation images, sketches, masks derived from both acquisition methods, and the captions, will be made available to researchers upon acceptance of this paper. Also, we will open-source the pre-trained weights files for DiffStroke.

## D More Editing Results

We provide some additional, high-resolution facial and natural image manipulation results as shown in Fig. 11 and Fig. 12, respectively.

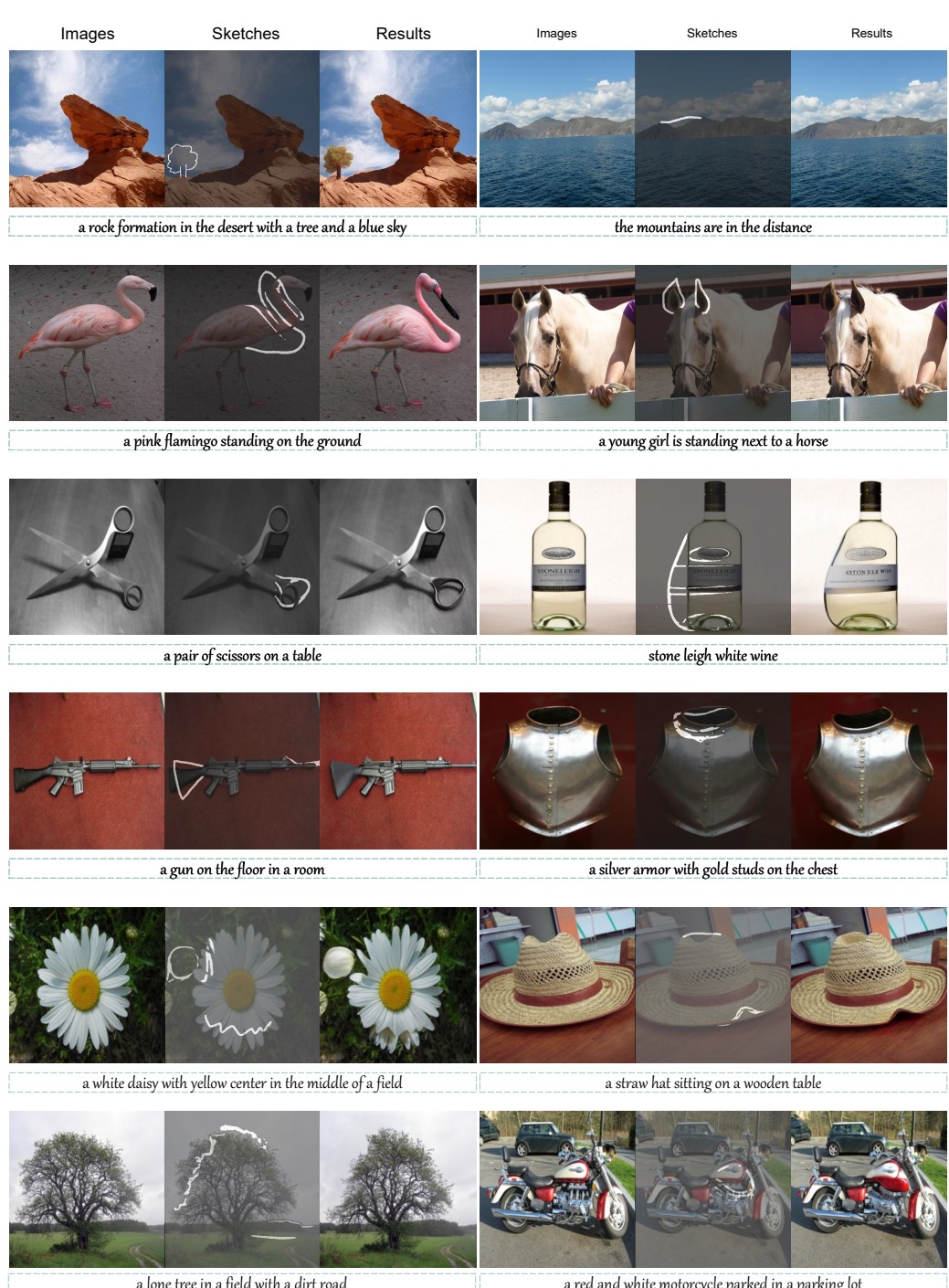

Figure 11: More examples of natural image manipulation by StrokeDiff.

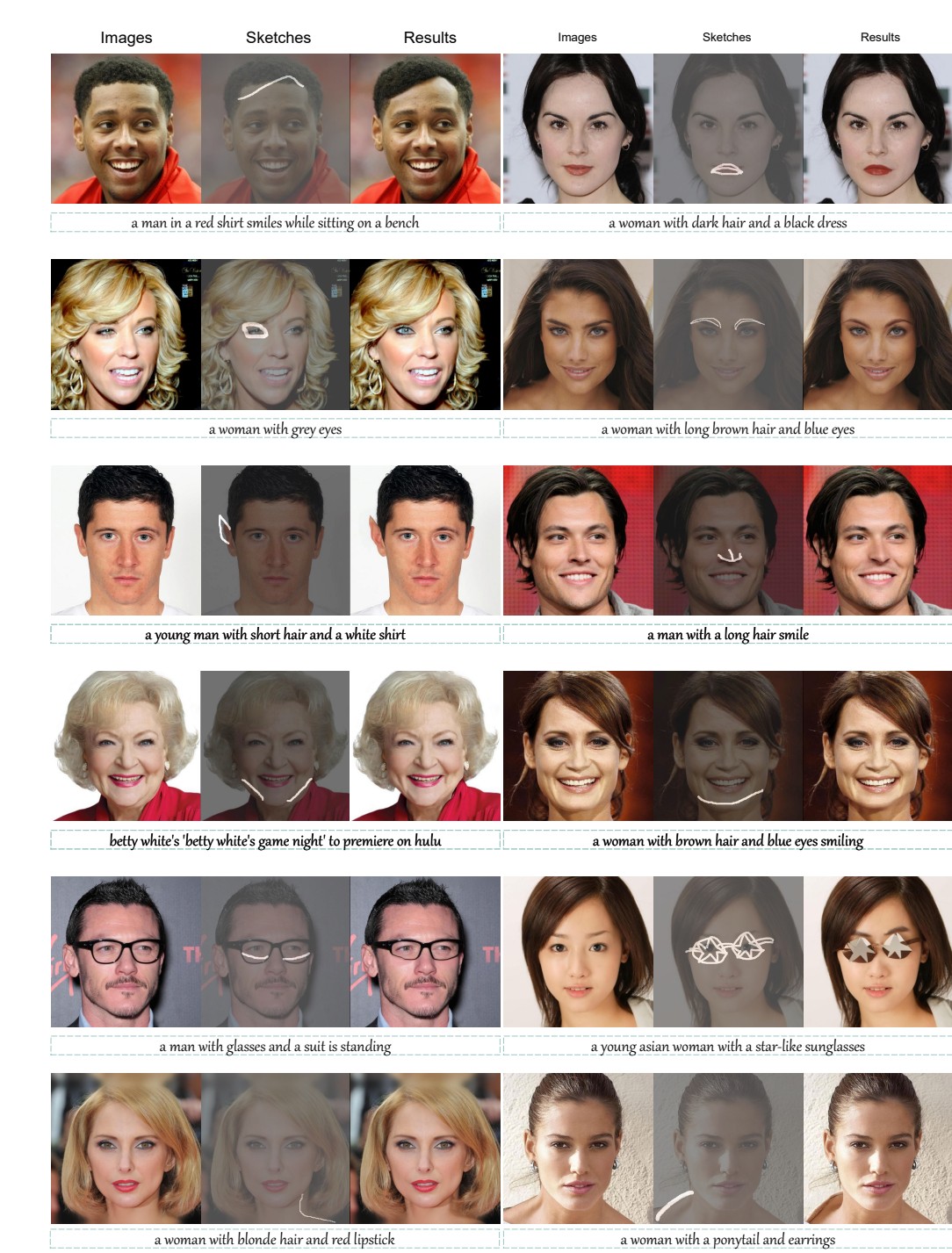

Figure 12: More examples of facial image manipulation by StrokeDiff.

## E  FAILURE CASES

Although our method has shown effectiveness in image editing, DiffStroke is still limited in some scenarios. Fig. 13 provides some failure cases. We observe that sketches with the same semantics as the objects to be edited but far away may not produce accurate masks, e.g., the second row in Fig. 13. In certain instances, although DiffStroke is capable of producing results that correspond

to the specified line control conditions, they do not meet the expectations typically associated with human performance. To illustrate, in the case of the facial image situated on the left side of the third line in Fig. 13, our objective is to reveal the left side of her forehead. However, the resulting image exhibits alterations in the details of the bangs, which are modified to align with the shape of the sketch. Sometimes the color of local details may be difficult to control accurately, such as the eyes of a seagull. To make the results of mask-free image editing using sketches consistent with human behavior, subsequent research might try to introduce information about human habits to guide the process of generation.

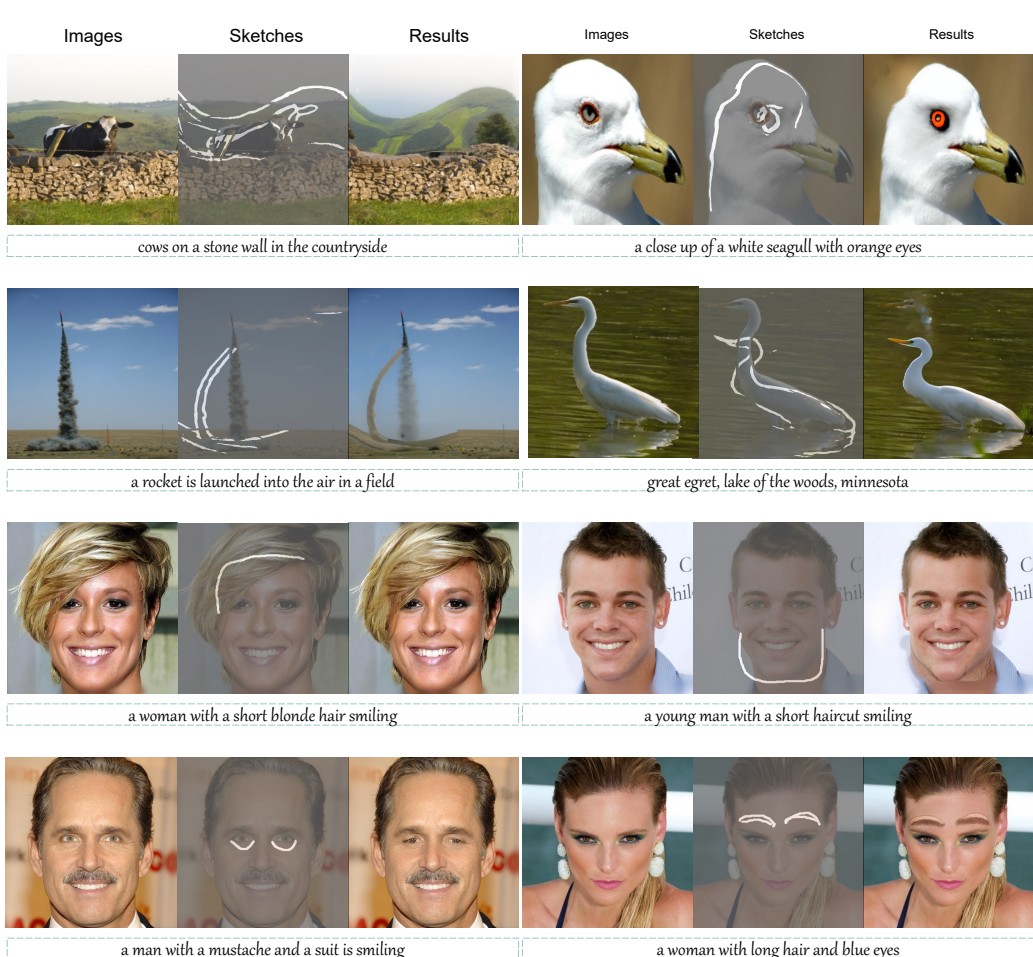

Figure 13: Failure cases of image manipulation by StrokeDiff.

# F PRELIMINARY

In this section, we provide preliminary knowledge about the Denoising Diffusion Probabilistic Models (DDPM) (Ho et al., 2020) and Denoising Diffusion Implicit Models (DDIM) (Song et al., 2021a).

## F.1 DDPM

DDPM is a generative model that aims to approximate the real data distribution $q_{data}(\boldsymbol{x}_0)$ and sample data from it. The DDPM consists of a forward process and a backward process. In the **forward process**, noise is gradually injected into the data $\boldsymbol{x}_0 \sim q_{data}(\boldsymbol{x}_0)$, which generates a series of the middle states $\boldsymbol{x}_1, \boldsymbol{x}_2, ..., \boldsymbol{x}_T$, to transform the data distribution into a simple distribution (i.e.,

Gaussian distribution). The process can be formalized as a Markov chain with Gaussian transitions:

$$q(\boldsymbol{x}_{1:T}|\boldsymbol{x}_0) = q(\boldsymbol{x}_0)\prod_{t=1}^{T} q(\boldsymbol{x}_t|\boldsymbol{x}_{t-1}),$$

$$q(\boldsymbol{x}_t|\boldsymbol{x}_{t-1}) = \mathcal{N}(\boldsymbol{x}_t; \sqrt{1-\beta_t}\boldsymbol{x}_{t-1}, \beta_t\boldsymbol{I}),$$

(12)

where $\beta_t \in (0,1)$ represents the noise schedule at time $t$.

The objective of the **backward process** is to reconstruct the data from a Gaussian noise $\boldsymbol{x}_T \sim \mathcal{N}(0,\boldsymbol{I})$ by sampling from $q(\boldsymbol{x}_{t-1}|\boldsymbol{x}_t)$ step by step. Since it's difficult to estimate the distribution $q(\boldsymbol{x}_{t-1}|\boldsymbol{x}_t)$ which is depended on the intractable distribution $q(\boldsymbol{x}_0)$, a neural network $p_\theta(\boldsymbol{x}_{t-1}|\boldsymbol{x}_t)$ is trained to approximate the distribution $q(\boldsymbol{x}_{t-1}|\boldsymbol{x}_t, \boldsymbol{x}_0)$ (a Gaussian distribution). This can be formalized as follows:

$$p_\theta(\boldsymbol{x}_{t-1}|\boldsymbol{x}_t) = \mathcal{N}(x_{t-1}; \boldsymbol{\mu}_\theta(\boldsymbol{x}_t,t), \boldsymbol{\Sigma}_\theta(\boldsymbol{x}_t,t)),$$

(13)

where $\boldsymbol{\mu}_\theta(\boldsymbol{x}_t,t)$ and $\boldsymbol{\Sigma}_\theta(\boldsymbol{x}_t,t))$ are the predicted mean and variance, respectively. The learning objective for diffusion model is derived by considering the variational lower bound,

$$
\begin{aligned}
\mathbb{E}\left[-\log p_\theta(\boldsymbol{x}_0)\right] &\leq \mathbb{E}_q\left[-\log\frac{p_\theta(\boldsymbol{x}_{0:T})}{q(\boldsymbol{x}_{1:T}|\boldsymbol{x}_0)}\right] \\
&= \mathbb{E}_q\left[-\log p(\boldsymbol{x}_T) - \sum_{t\geq 1}\log\frac{p_\theta(\boldsymbol{x}_{t-1}|\boldsymbol{x}_t)}{q(\boldsymbol{x}_t|\boldsymbol{x}_{t-1})}\right] \\
&= \mathbb{E}_q\left[\underbrace{D_{\mathrm{KL}}(q(\boldsymbol{x}_T|\boldsymbol{x}_0)\|p(\boldsymbol{x}_T))}_{L_T}\right. \\
&\quad \left.+ \sum_{t>1}\underbrace{D_{\mathrm{KL}}(q(\boldsymbol{x}_{t-1}|\boldsymbol{x}_t,\boldsymbol{x}_0)\|p_\theta(\boldsymbol{x}_{t-1}|\boldsymbol{x}_t))}_{L_{t-1}}\underbrace{-\log p_\theta(\boldsymbol{x}_0|\boldsymbol{x}_1)}_{L_0}\right].
\end{aligned}
$$

(14)

Instead of estimating $\boldsymbol{\mu}_\theta(\boldsymbol{x}_t,t)$ directly, DDPM utilize an approximator $\boldsymbol{\epsilon}_\theta(\boldsymbol{x}_t,t)$ to predict the noise $\epsilon$ that was introduced to $\boldsymbol{x}_0$ obtain $\boldsymbol{x}_t$. The training objective is as follows:

$$
\begin{aligned}
&\min_\theta \mathbb{E}_q D_{KL}(q(\boldsymbol{x}_{t-1}|\boldsymbol{x}_t,\boldsymbol{x}_0)\|p_\theta(\boldsymbol{x}_{t-1}|\boldsymbol{x}_t)) \\
&= \min_\theta \mathbb{E}_{\boldsymbol{x}_0,\epsilon\sim\mathcal{N}(0,\boldsymbol{I}),t\sim Uniform(1,T)}\|\epsilon - \boldsymbol{\epsilon}_\theta(\boldsymbol{x}_t,t)\|_2^2.
\end{aligned}
$$

(15)

Then $\boldsymbol{\mu}_\theta(\boldsymbol{x}_t,t)$ can be derived using Bayes' theorem,

$$\boldsymbol{\mu}_\theta(\boldsymbol{x}_t,t) = \frac{1}{\alpha_t}(\boldsymbol{x}_t - \frac{\beta_t}{\sqrt{1-\bar{\alpha}_t}}\boldsymbol{\epsilon}_\theta(\boldsymbol{x}_{,t},t)),$$

(16)

where where $\alpha_t = 1 - \beta_t$ and $\bar{\alpha}_t = \prod_{i=1}^t \alpha_i$. In the inference stage, the sampled noise $\boldsymbol{x}_T \sim \mathcal{N}(0,\boldsymbol{I})$ is repeatedly denoised by eq. 13 until $t = 0$. More details can be accessed in (Ho et al., 2020; Song et al., 2021b).

### F.2 DDIM

To improve the sampling efficiency of DDPM (Ho et al., 2020), DDIM (Song et al., 2021a) breaks the Markov property of the DDPM reverse process. The researchers found that the forward process, if it can be in the following form:

$$q_\sigma(\boldsymbol{x}_t|\boldsymbol{x}_0) = \mathcal{N}(\boldsymbol{x}_t; \sqrt{\bar{\alpha}_t}\boldsymbol{x}_0, (1-\bar{\alpha}_t)\boldsymbol{I}),$$

$$q_\sigma(\boldsymbol{x}_{1:T}|\boldsymbol{x}_0) = q_\sigma(\boldsymbol{x}_T|\boldsymbol{x}_0)\prod_{t=2}^{T} q_\sigma(\boldsymbol{x}_{t-1}|\boldsymbol{x}_t,\boldsymbol{x}_0),$$

(17)

the constraint of Markov property can be eliminated. Then they derive that

$$q_\sigma(\boldsymbol{x}_{t-1}|\boldsymbol{x}_t,\boldsymbol{x}_0) = \mathcal{N}(\sqrt{\bar{\alpha}_{t-1}}\boldsymbol{x}_0 + \sqrt{1-\bar{\alpha}_{t-1}-\sigma_t^2}\cdot\frac{\boldsymbol{x}_t - \sqrt{\bar{\alpha}_t}\boldsymbol{x}_0}{\sqrt{1-\bar{\alpha}_t}}, \sigma_t^2\boldsymbol{I}),$$

(18)

where $t \geq 2$ and $q_\sigma(\boldsymbol{x}_T|\boldsymbol{x}_0) = \mathcal{N}(\boldsymbol{x}_T; \sqrt{\bar{\alpha}_t}\boldsymbol{x}_0, (1-\bar{\alpha}_t)\boldsymbol{I})$. With the utilization of Bayes' rule, the forward process in DDIM can be expressed as

$$q_\sigma(\boldsymbol{x}_t|\boldsymbol{x}_{t-1}, \boldsymbol{x}_0) = \frac{q_\sigma(\boldsymbol{x}_{t-1}|\boldsymbol{x}_t, \boldsymbol{x}_0)q_\sigma(\boldsymbol{x}_t|\boldsymbol{x}_0)}{q_\sigma(\boldsymbol{x}_{t-1}|\boldsymbol{x}_0)}, \tag{19}$$

that $\boldsymbol{x}_t$ is no longer dependent on $\boldsymbol{x}_{t-1}$ but also $\boldsymbol{x}_0$. Finally, the denoising step is derived as follows:

$$\boldsymbol{x}_{t-1} = \sqrt{\alpha_{t-1}} \underbrace{\left(\frac{\boldsymbol{x}_t - \sqrt{1-\bar{\alpha}_t}\epsilon_\theta^{(t)}(\boldsymbol{x}_t)}{\sqrt{\bar{\alpha}_t}}\right)}_{\text{"predicted } \boldsymbol{x}_0\text{"}} + \underbrace{\sqrt{1-\bar{\alpha}_{t-1}-\sigma_t^2} \cdot \epsilon_\theta^{(t)}(\boldsymbol{x}_t)}_{\text{"direction pointing to } \boldsymbol{x}_t\text{"}} + \underbrace{\sigma_t\epsilon_t}_{\text{random noise}}, \tag{20}$$

where the variance $\sigma_t^2$ is defined as $\sigma_t^2 = \eta \cdot \tilde{\beta}_t = \eta\sqrt{(1-\bar{\alpha}_{t-1})/(1-\bar{\alpha}_t)}\sqrt{1-\bar{\alpha}_t/\bar{\alpha}_{t-1}}$. In the case where $\eta = 1$, the denoising process is consistent with that of DDPM. Conversely, when $\eta = 0$, the sampling process becomes deterministic, thereby resulting in the DDIM step

$$\boldsymbol{x}_{t-1} = \sqrt{\alpha_{t-1}} \left(\frac{\boldsymbol{x}_t - \sqrt{1-\bar{\alpha}_t}\epsilon_\theta^{(t)}(\boldsymbol{x}_t)}{\sqrt{\bar{\alpha}_t}}\right) + \sqrt{1-\bar{\alpha}_{t-1}} \cdot \epsilon_\theta^{(t)}(\boldsymbol{x}_t). \tag{21}$$

### F.3 DDIM INVERSION (REVERSE DDIM STEP)

To generate images in a controllable manner by GANs (Goodfellow et al., 2014), a manipulable encoding $\boldsymbol{z}$ is frequently obtained by utilizing the inverse mapping $\boldsymbol{z} = G^{-1}(\boldsymbol{x})$ of the generative process $\boldsymbol{x} = G(\boldsymbol{z})$. For the diffusion model, intuitively we can correspond the forward process to $\boldsymbol{z} = G^{-1}(\boldsymbol{x})$ and the reverse process to $\boldsymbol{x} = G(\boldsymbol{z})$. However, in DDPM (Ho et al., 2020), the two processes are not reversible due to the introduction of random noise at each sampling step, which results in $\boldsymbol{x}_T$ not being in a one-to-one correspondence with $\boldsymbol{x}_0$. Fortunately, DDIM (Song et al., 2021a) eliminates the ambiguity associated with the sampling process, thereby facilitating the implementation of image manipulation techniques based on diffusion models. Given the data $\boldsymbol{x}_0$, we can derive the equation from the eq. 21,

$$\boldsymbol{x}_t = \sqrt{\frac{\bar{\alpha}_t}{\bar{\alpha}_{t-1}}}\boldsymbol{x}_{t-1} + \sqrt{\bar{\alpha}_t}\left(\sqrt{\frac{1}{\bar{\alpha}_t}-1} - \sqrt{\frac{1}{\bar{\alpha}_{t-1}}-1}\right)\epsilon_\theta(\boldsymbol{x}_t, t), \tag{22}$$

which is applied to obtain the state $\boldsymbol{x}_1, \boldsymbol{x}_2, ..., \boldsymbol{x}_T$. Nevertheless, the term $\epsilon_\theta(\boldsymbol{x}_t, t)$ is not able to be calculated directly, $\epsilon_\theta(\boldsymbol{x}_{t-1}, t-1)$ is considered for approximating it. In the case of sufficiently small time step intervals, $\epsilon_\theta(\boldsymbol{x}_t, t) \approx \epsilon_\theta(\boldsymbol{x}_{t-1}, t-1)$ is believed to hold. Finally, the reverse DDIM step is as follows:

$$\boldsymbol{x}_t = \sqrt{\frac{\bar{\alpha}_t}{\bar{\alpha}_{t-1}}}\boldsymbol{x}_{t-1} + \sqrt{\bar{\alpha}_t}\left(\sqrt{\frac{1}{\bar{\alpha}_t}-1} - \sqrt{\frac{1}{\bar{\alpha}_{t-1}}-1}\right)\epsilon_\theta(\boldsymbol{x}_t-1, t-1). \tag{23}$$

