# OpenReview forum: "DiffStroke: High-Quality Mask-free Image Manipulation with Partial Sketches"
_ICLR.cc/2025/Conference — ICLR 2025 Conference Withdrawn Submission_

### Official Review · Reviewer_T7Jx · 2024-10-18

**Soundness:** 3
**Presentation:** 3
**Contribution:** 2
**Rating:** 5
**Confidence:** 4

**Summary:**

This paper proposes a novel plug-and-play module, namely the Image-Stroke Fusion (ISF) module, for mask-free image editing using strokes. The ISF module contains a simple concatenation of the source image features extracted by the T2I-adapter and the sketch information, and its shallowest layer is used for mask prediction. Experimental results are visually pleasing.

**Strengths:**

(+) The paper is well-written and easy to follow.

(+) The results are visually pleasing.

**Weaknesses:**

(-) My major concern is that the technical contributions are thin and the methodology is a bit too straightforward. The proposed method relies heavily on T2I-adapter and the proposed Image-Stroke Fusion (ISF) module does not demonstrate many architectual contributions as it contains a simple concatenation of the source image features extracted by the T2I-adapter and the sketch information, and the mask prediction branch is also straightforward. There are not many novel insights here as well (Why the ISF can improve quality? What is the merit of the proposed mask estimation strategy compared to existing ones?). The editing through masking is also a common technique in diffusion-based models.

(-) The data preparation part follows a similar strategy in previous works and also limits the scope of sketch editing to shape changes.

(-) The paper claims that the module is plug-and-play but there are only experiments with T2I-adapter, experiments on several other models are required to justify this claim.

**Questions:**

Please see weaknesses above.

---

### Official Review · Reviewer_Z4ja · 2024-10-21

**Soundness:** 3
**Presentation:** 3
**Contribution:** 2
**Rating:** 5
**Confidence:** 4

**Summary:**

This paper presents DiffStroke, a mask-free method for high-quality image editing using only partial sketches. The proposed approach includes a trainable plug-and-play image-stroke fusion (ISF) module and a mask estimation module aimed at addressing the limitations of previous techniques. Experimental results demonstrate that DiffStroke outperforms existing methods in both simple and complex stroke-based editing tasks.

**Strengths:**

- The motivation is clear, the experiments are thorough, and the presentation is clear and easy to understand.
- Both qualitative and quantitative comparisons with state-of-the-art methods demonstrate the effectiveness of the proposed methods.
- There are ablation studies that confirm the efficacy of the proposed module.

**Weaknesses:**

- Experiments from the ablation studies show that providing masks from estimation is important. Creating a mask is not as difficult as drawing sketches; for example, it can be easily obtained from users by scribbling. Masks can also be derived from sketches by identifying the outline boundaries. What would be the differences between these approaches? It would be beneficial to compare the proposed method by modifying it to accept both masks and sketches as inputs, using the outline boundaries directly obtained from the given sketch as the mask.

- The method is built upon a sketch-based T2I adapter with modifications to the feature infusion. Compared to the T2I adapter, the form of information fusion does not seem fundamentally different; it simply passes the features through transformer layers to fuse them and uses an additional MLP for mask estimation. Could you elaborate on the differences and possibly conduct ablation studies on the structure of the ISF blocks? It would be helpful to see a comparison with minimal changes made by adapting the idea based on the current T2I adapter structure.

- The paper did not provide diverse generation results or failure cases. It would be helpful to show diverse results with the same sketch and prompt, the same sketch with different prompts, or the same prompt with different sketches. Additionally, what would happen without prompts? In the provided results in Figure 6, the prompt appears quite lengthy, making the editing process less user-friendly.

- Some edited results appear less natural than those from the T2I adapter or ControlNet. It would be beneficial to conduct user studies comparing the results from different methods in terms of consistency and naturalness.

**Questions:**

Please find the questions in the Weaknesses.

**Details Of Ethics Concerns:**

This technology can be used for creating deepfakes.

---

### Official Review · Reviewer_6iAm · 2024-10-29

**Soundness:** 2
**Presentation:** 3
**Contribution:** 3
**Rating:** 5
**Confidence:** 4

**Summary:**

This paper proposes a DiffStroke framework for mask-free image editing with sketches. It designs an image-stroke fusion module to fuse the features of original images and stoke images to predict the editing mask and guide the diffusion-based conditional image generation process. This method is plug-and-play and shows good performance over other mask-based or mask-free image editing methods. The main contribution is that the proposed method is make-free, which can save many users’ efforts during editing.

**Strengths:**

**Originality** This methods propose a ISF block, to fuse the features of original images and stoke images to predict the editing mask and guide the diffusion-based conditional image generation process. With this design, the users are free from drawing masks, which make this method user-friendly. I think this task is interesting and valuable.

**Performance** Generally, I found this method perform good. Even without masks, this method outperforms many other mask-based methods, and show more flexibilities through more precise mask prediction.

**Weaknesses:**

**Experimental results** Although most of the results look reasonable, some cases have limitations. For example, in the Fig. 4, in the last case (church), the unedited part in the left of the sketches is incorrectly edited into a tree. It seems that the predicted mask is not precise enough and would harm the content of the original image in the unedited region.

Then, another limitation of this paper, as also discussed in the paper, is that this paper is designed to only edit the shape of the objects under a small distortion. To me, the proposed methods handle some image morphing or image warping operations, lacking the generative abilities of the original diffusion models. It cannot handle large region changes such as replacing some objects or creating some new objects. I believe this greatly limits the application scenarios of the method.

In addition, the proposed method is claimed to be `plug-and-play` (Line 120). However, I didn’t find validations about this.

**Questions:**

**Experimental results**

1.	In Fig. 4 and Fig. 5, the predicted masks are suggested to be given.
2.	I wonder how the prompts affect the output? Given the same image and sketch, using different prompts will generate the same image or diverse image. Will wrong or imprecise prompts fail the method?
3.	How to calculate the mask without ISF in the ablation study of Fig. 6?
4.	Although the authors discuss the reason why SketchEdit shows poor results on human faces, it is still valuable to conduct comparisons with SketchEdit on the official images provided by SketchEdit.
5.	The proposed method is claimed to be `plug-and-play`. However, there is no validation about this. For example, what is the performance when applied to controlnet?

Some unclear details or minor issues:

1.	Line 258, why use $S_{tar}-S_{src}$? This will lead to some -255 values in the images? Should it be $max(0, S_{tar}-S_{src})$?
2.	Line 272, $h_i^{src}$ and $h_i^s$ should be $\mathbf{h_i}^{src}$ and $\mathbf{h_i}^{s}$
3.	Line 286 and Line 295, MLP is misleading, since the implementation of $f_{MLP}(.)$ is a one-layer CNN. It should be $f_{CNN}(.)$
4.	Line 318 and Line 373, `Eq. 3` should be `Eq. (3)`
5.	Line 327, it is unclear why t=273 is used? In what part of the algorithm it is used? Only for extracting the features from $z_{src}$? Why not using the same noise level as the main network for feature fusion?
6.	Line 335, `mask mask` should be `mask`
7.	Line 466, `the metrics of the metrics`

---

### Official Review · Reviewer_f2wQ · 2024-11-11

**Soundness:** 3
**Presentation:** 3
**Contribution:** 2
**Rating:** 5
**Confidence:** 4

**Summary:**

This manuscript presents a method, called DiffStroke, for image editing based on partial sketch inputs using diffusion models. The method consists of two components, the trainable plug-and-play Image-Stroke Fusion (ISF) module and a mask estimator. The ISF modules fuse the sketch input encodings with the source image features. The mask estimator estimates a mask based on the input sketch to prevent alternation in irrelevant areas. Experimental results demonstrate that the proposed method outperforms previous methods.

**Strengths:**

1. The method is based on diffusion models, which are shown to produce higher-quality and more diverse images than GANs. Therefore, it is unsurprising that the proposed method outperforms the previous approaches based on GANs.
2. The method can edit an image based on partial input sketches without masks. It is more user-friendly than ControlNet and inpainting.
3. The model is trained using hand-drawn sketches, reducing the gap between training and test-time conditions.

**Weaknesses:**

1. The main weakness is the limited novelty. There are multiple existing mask-free sketch-based image editing methods. The main difference between the proposed method and the existing methods is that the proposed method uses the diffusion model while the existing methods are based on GANs. This contribution is not significant enough for a top conference like ICLR.
2. The proposed method uses free-form deformation to generate training data, limiting the editing capability to simple deformations and resulting in visually unpleasant images. Many result images shown in the paper are not aesthetically pleasing, for instance, the face in Figure 6 and the mug in  Figure 2 do not look natural.
3. The proposed method is not useful in practice. The technique employs a heavy text-based diffusion model while only being able to produce simple deformations. Users can get a similar effect with PhotoShop in a few simple steps

**Questions:**

1. Related to the weakness (2), would swapping I_src and I_tar, i.e. using the deformed image and sketch as input with the original image as ground truth, give better results?
2. How are the hyperparameters 2.5, 0.25, and 273 in Equation 10 decided? How do the choices on these values affect the results?

---

### Note · Authors · 2024-11-14

**Comment:**

We thank the reviewers for their suggestions and will improve the paper in subsequent editions.

**Withdrawal Confirmation:**

I have read and agree with the venue's withdrawal policy on behalf of myself and my co-authors.